# Vertical transmission of Dengue virus type-3 and metagenomic virome profiles of *Aedes aegypti* mosquitoes collected in Kisumu, Kenya

Tabitha Wanjiru[1,2,3]*, Wallace Bulimo[2], Solomon Langat[2], Johnson Kinyua[3], Nicholas Odemba[1], Santos Yalwala[1], David Oullo[1], Richard Ochieng[1], Francis Ngere[1], Gladys Kerich[1], Janet Ambale[1], Eunice Achieng[1], David Abuom[1], Timothy Egbo[1], Jaree Johnson[4], Elly Ojwang[1], John Eads[1], Eric Garges[1], Fredrick Eyase[1,2]

**1** Department of Emerging Infectious Diseases, Walter Reed Army Institute of Research-Africa, Nairobi, Kenya, **2** Centre for Virus Research, Kenya Medical Research Institute, Nairobi, Kenya, **3** Department of Biochemistry, Jomo Kenyatta University of Agriculture and Technology (JKUAT), Nairobi, Kenya, **4** Department of Entomology, Walter Reed Army Institute of Research, Silver Spring, Maryland, United States of America

* wanjirutabs@gmail.com

## Abstract

*Aedes aegypti* is the main vector of several arboviruses including chikungunya, dengue, yellow fever and Zika. Beyond arboviruses, *Aedes aegypti* harbours insect-specific viruses (ISVs), which can modulate mosquito's ability to transmit diseases by interfering with viral processes and triggering immune responses. Both arboviruses and ISVs can be transmitted vertically, where viruses are passed from parent to offspring. The lack of systematic molecular and entomological surveillance, has left the diversity of viruses in local *Aedes aegypti* populations largely unexplored. This study aimed to characterize the viromes of *Aedes aegypti* mosquitoes from Kisumu, Kenya, focusing on viral diversity. Immature larvae and pupae were collected from Jua Kali area in Kisumu, reared into adults, and subjected to viral isolation by cell culture and metagenomic next-generation sequencing. RNA extraction, library preparation, and Illumina MiSeq sequencing were performed on CPE positive pools and metagenomic superpools. Initial data analysis was conducted using the CZ-ID platform, with quality control applied using PrinseqLite v0.20.4 to filter low-quality reads and remove adapters. De novo sequence assembly was performed with MEGAHIT v1.2.9, followed by BLAST analysis. Phylogenetic relationships were analyzed using the Maximum Likelihood method. A total of 2,142 female *Aedes aegypti*, grouped into 86 pools and 4 superpools, were analyzed using cell culture and metagenomic next-generation sequencing respectively. Dengue virus type-3 was detected in one of the 86 pool. Additionally, a variety of ISVs were identified, including Iflaviruses related to Tesano Aedes Iflavirus (TeAV), Armigeres Iflavirus, and Negeviruses related to Rabai Virus. An unclassified virus closely related to Korle-Bu Aedes virus was also detected. Our study provides insights into the viral diversity within

**Data availability statement:** All data is available at GenBank under accession numbers PV419618, PP590324, PP590325, PP590327 and PP590328.

**Funding:** This work was funded by the Armed Forces Health Surveillance Branch (AFHSB) and its Global Emerging Infections Surveillance (GEIS) Section, FY2022 ProMIS ID: P0116_22_KY and FY2023 ProMIS ID P0094_23_KY. The funders had a role on validation of the project, administration and decision to publish.

**Competing interests:** The authors have declared that no competing interests exist.

**Abbreviations:** CHIKV, Chikungunya virus; CPE, Cytopathic effect; nt, nucleotides; DENV, Dengue virus; FBS, Fetal bovine serum; MEM, Minimum essential medium; VT, Vertical Transmission.

*Aedes aegypti* mosquitoes in Kisumu and evidence of natural vertical transmission, specifically transovarial transmission of dengue virus type-3. Ongoing research is imperative to unravel vertical transmission mechanisms and subtleties governing ISV-arbovirus interactions across diverse environmental settings.

## Introduction

*Aedes aegypti* is a known vector for numerous arboviruses, including yellow fever, dengue fever, chikungunya and Zika viruses widely distributed in tropical and sub-tropical regions around the world [1]. It is a day-biting mosquito, and often feeds on multiple hosts during a single gonotrophic cycle [2]. It is anthropophilic and is well established in the ever-expanding metropolitan environment [3]. *Aedes aegypti* exists in two forms [4]: the anthropophilic, light-coloured *Aedes aegypti aegypti (Aaa)* and the dark-colored, sylvatic *Aedes aegypti formosus (Aaf)* [5]. In contrast to *Aedes aegypti aegypti*, which is commonly found in urban areas, *Aedes aegypti formosus* has been found in vegetated environments in western Kenya, close to Kisumu City, as well as in the Kakamega forest [4]. Notably, both forms have been found to exist along the Kenyan coast at Rabai [5]. More *Aedes aegypti* are detected during long rains than during the dry season and short rains [6]. They are also found both inside houses and outdoors with the outdoor numbers increasing significantly in the afternoons than in the morning hours [7]. *Aedes aegypti* populations are up to three times higher in urban areas than in rural areas [8]. Water-holding containers, particularly buckets, drums, tires, pots, and jerrycans in the outdoors, have been shown to provide a large number of breeding sites for *Aedes aegypti* in urban environments [9]. These traits have significant consequences for the potential spread of arboviral infections, as well as the design of surveillance and control strategies. It is apparent that arbovirus outbreaks are more likely to occur in urban than in rural regions, however source reduction measures aimed at outdoor and indoor containers could be a cost-effective strategy to reduce outbreaks [9,10] A previous study in Kenya has shown that water holding containers found outdoors in most Kenyan towns provide 75% of dengue vector breeding sites [11]. *Aedes aegypti* causes dengue fever after infection with any of the four serotypes of dengue virus (DENV-1, DENV-2, DENV-3, DENV-4) [12,13]. The symptoms include fever, headache, retro-orbital eye pain, myalgia, arthralgia, mild hemorrhagic signs, and a rash. Although most dengue patients in endemic areas recover within a week, 5–10% develop severe dengue, which manifests as dengue hemorrhagic fever (DHF) and dengue shock syndrome (DSS) [14,15]. The disease is widespread in more than 125 countries around the world, primarily in Asia, the Americas, and Africa, where roughly 3.6 billion people are currently at risk [16]. In Kenya dengue outbreaks have been reported over time. The first recorded case was identified in a Canadian tourist who had visited Malindi in 1982. Subsequent serological studies confirmed the presence of Dengue virus serotype 2 (DENV-2) among residents and visitors in these areas [17]. It was thought to have spread from the outbreak that had occurred in the Seychelles between 1977 and 1979 [18]. Then after almost 30 years, dengue outbreaks occurred in Mandera

in northern Kenya in 2011 where 30 cases were confirmed positive for dengue virus type 3 [19] and subsequently in Mombasa city along the Kenyan coast in 2013–2014 where 155 cases were confirmed to have DENV infection [11] Consequently, sporadic outbreaks have been reported mainly in the North-Eastern Kenya and Coastal Kenya in May 2017 where 295 cases were confirmed positive [20,21]. The most recent outbreak occurred in 2021 where 553 cases were reported in the coastal region [22,23].

Other than arboviruses, *Aedes aegypti* harbors Insect-Specific Viruses (ISVs) that do not directly infect vertebrates. Some of these ISVs are unique to *Aedes aegypti* species and they include *Aedes aegypti Densovirus* (AeDNV), a DNA virus from the Parvoviridae family that impacts larval development and survival, as well as Aedes Flavivirus (AEFV) and Kamiti River Virus (KRV), both of which are commonly found in *Aedes aegypti* populations and may influence infection dynamics within the mosquito [24,25]. Mounting evidence indicate that ISVs interact with arboviruses and may suppress their replication [26]. Recent studies have shown that ISVs, such as cell fusing agent virus (CFAV) [27], Nhumirim virus (NHUV), and Palm Creek virus (PCV) [28], have demonstrated the ability to reduce viral loads of vertebrate pathogenic flaviviruses, like West Nile virus (WNV), Zika virus (ZIKV), dengue virus (DENV), and Japanese (JEV) and St. Louis encephalitis (SLEV) viruses [29]. Similarly, the insect-specific alphavirus Eilat virus (EILV) [30] was shown to reduce replication of the pathogenic alphaviruses chikungunya virus (CHIKV), Sindbis virus (SINV), and eastern (EEEV), western (WEEV), and Venezuelan equine encephalitis (VEEV) viruses in cell culture [31]. Other ISVs have shown to suppress the antiviral RNA interference (RNAi) response, or enhance the transcription of host factors [32]. The mechanism(s) of interference remains the subject of ongoing research. Due to arbovirus surveillance programs, novel insect-specific viruses (ISVs) have been discovered in mosquitoes based on advances in sequencing technology and a growing awareness of the virome. As such, the ability of ISVs to interfere with pathogenic arboviruses and be maintained in vector populations calls for further investigation.

Viral transmission can be either horizontal (human-mosquito-human cycle) or vertical (from parents to their offspring) [32]. Horizontal transmission is largely considered the major route by which the virus remains in an area. However, Vertical transmission has been suggested as a mechanism that ensures conservation of the virus during conditions that would be adverse for horizontal transmission (i.e., harsh winters and interepidemic stages). It also represent the only known route for ISVs transmission in nature [33]. Vertical transmission may occur by either transovarial transmission, in which the virus infects germinal tissues of the female including oocytes or through trans-ovum transmission, which occurs during fertilization or by viral infection of the fully intact mature eggs during oviposition [32]. Mosquito eggs are capable of surviving in the environment under adverse conditions for over one year [34]. The eggs may undergo periods of either diapausing or desiccation and could provide a reservoir of arboviruses capable of persisting through seasons of low adult vector abundance [35]. Recent findings indicate that vectors are being born already infected and able to transmit the virus at the beginning of epidemics [32]. This ability serves as a mechanism of virus maintenance between interepidemic periods, thereby favouring arbovirus transmission to susceptible hosts and subsequent dengue outbreaks in the studied neighbourhood.

More recently, vertical transmission of dengue virus has been documented in the field, where different serotypes of dengue viruses in larvae were detected in different areas of the world [36,37]. In Kenya, recent studies have shown evidence of vertical transmission during both outbreak and interepidemic periods, highlighting the potential of mosquitoes to act as reservoirs for the virus, which could contribute to the resurgence of the disease even after periods of low transmission [10,38]. Presence of viruses in mosquitoes collected in the field allows for the detection of epidemics from six to eight weeks in advance of their onset. Mosquitoes and larvae may be infected by vertical transmission and maintain the virus in nature, therefore, monitoring of *Aedes* larvae for arboviruses could be used to predict epidemics [39]. However, transmission of arboviruses is dependent on several parameters such as virus multiplication dynamics, the ecology and behaviour of *Aedes aegypti.* Transmission efficiency may be affected by extrinsic factors, e.g., rainfall, temperature and intrinsic factors, e.g., genetic factors [40].

Viral isolation by cell culture is a critical method for detecting arboviruses, offering direct evidence of infectious viruses and enabling their detailed characterization. Using cell lines like Vero (African green monkey kidney cells), the process involves inoculating samples onto susceptible cells, monitoring for cytopathic effects (CPE), and confirming viral presence through molecular or serological methods. This technique is essential for identifying active infections, discovering novel arboviruses and studying viral genetics and replication. It supports epidemiological surveillance, providing invaluable insights into arbovirus diversity, evolution, and transmission dynamics [41].

Metagenomic sequencing has exponentially increased the number of mosquito-borne viruses detected in the last couple of years and further provided fresh insight into the enormous complexity and variety of invertebrate RNA viruses [42]. A greater understanding of the virome in mosquito species could allow for a more accurate assessment of mosquito-borne disease risk, vector competence and mosquito management. At present, vector control is an important component of public health interventions to reduce the spread of mosquito-borne diseases. The surveillance of mosquitoes and mosquito-borne pathogens is a significant element of early warning, prevention, and control of infectious diseases. Within Western, Kenya, there has been no systematic molecular entomological surveillance for arboviruses and thus limited knowledge of the viral diversity. This study leveraged on cell culture for isolation of arboviruses and additionally, the unbiased high throughput metagenomic next-generation sequencing strategy, allowing comprehensive identification of all genetic material within a sample, to detect the virome of *Aedes aegypti* from Jua kali area in Kisumu.

## Materials and methods

### Ethical approval

Ethical approval was obtained from the Kenya Medical Research Institute (KEMRI) Scientific and Ethics Review Unit (SERU) under protocol number KEMRI/SERU/CCR/4702 and WRAIR# 3101. Permission to conduct the study was granted by the National Council for Science, Technology, and Innovation (NACOSTI).

### Study area

The study was carried out in Jua Kali, Kisumu west sub-county (Fig 1). The area lies between longitudes 33° 20' E and 35° 20' E and latitude 00° 20' South and 00° 50' South.

### Weather data

The prevailing weather data for Jua kali area during the sampling period was obtained from the Kenya Meteorological Department (Fig 2). Most of November, 2022 was relatively dry with the lowest mean Rainfall 3.9 mm and mean temperature of 17.7°C. However, there were instances of heavier rainfall, with the highest amount reaching 16.8 mm.

### Entomological investigation

Sampling of immature mosquitoes was conducted from 7th to 15th November 2022. Larvae and pupae were collected daily. A total of 58 containers consisting of 52 tyres, 4 cans/tins and 2 jerry cans were sampled.. Samples from each positive container were collected using ladles and pipettes or, in the case of jerry cans and cans/tins, the water was poured onto a white basin and the larvae or pupae picked. The samples were georeferenced to the sites where they were collected by geo-coding using a GPS, taken to the laboratory and reared into adults. The adult mosquitoes were immobilized by freezing at −20°C for 20 minutes, identified morphologically to species under a dissecting microscope using taxonomy keys, including Edwards (1941) [43], Harbach (1988) [44] and Jupp (1986) [45]. The identified mosquitoes were pooled in groups of 1–25 samples based on species, sex and collection date, and stored at −80 °C.

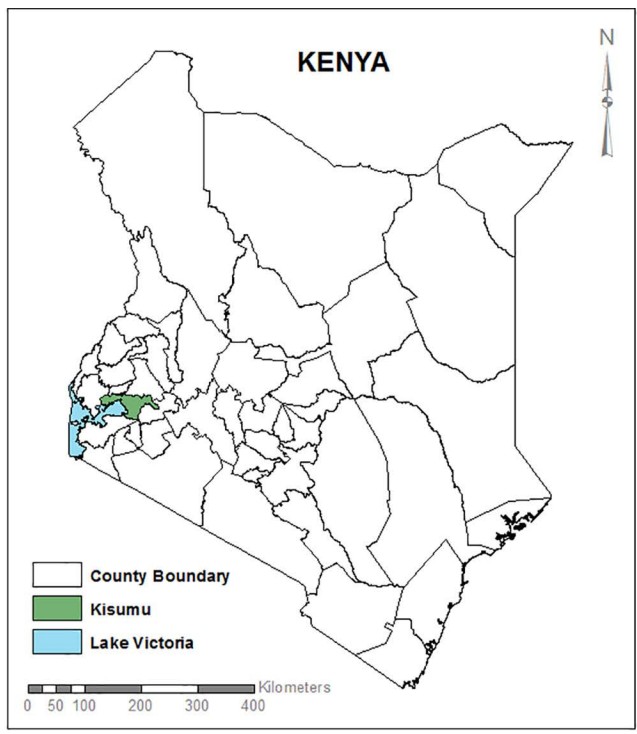

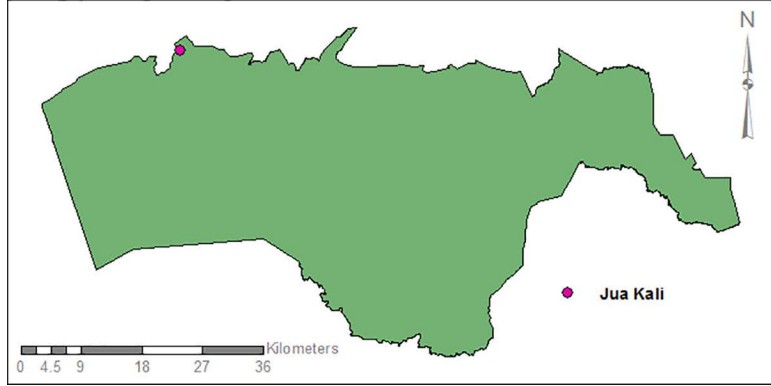

**Fig 1. The overall map of Kenya and Kisumu County showing the site where sampling was conducted.** Base maps, boundaries and shape files of Kenyan map and administrative boundaries of the County and Sub-county were derived from GADM data version 4.1 (https://gadm.org) and the maps were generated using ArcGIS Version 10.2.2 (http://desktop.arcgis.com/en/arcmap) advanced license) courtesy of Samuel Owaka.

## Mosquito sample preparation

Two thousand one hundred and forty-two (2,142) adult female *Aedes aegypti* mosquitoes were grouped into pools with each pool having a maximum of 25 and a minimum of 20 resulting in eighty-six (86) mosquito pools. The pools were homogenized (completely crushed) with zirconium beads (2.0 mm diameter) for 40 seconds using a Mini-Beadruptor-16 (Biospec, Bartlesville, OK, USA) in 1000 µL of homogenization media that consisted of minimum essential media supplemented with 15% fetal Bovine Serum (FBS) (Gibco by Life Technologies, Grand Island, NY, USA), 2% L-glutamine (Sigma, Aldrich) and 2% antibiotic/antimycotic (Gibco by Life Technologies, Grand Island, NY, USA.). Subsequently, the homogenate was centrifuged at 10,000 rpm for 10 minutes at 4°C using a benchtop centrifuge (Eppendorf, USA). The

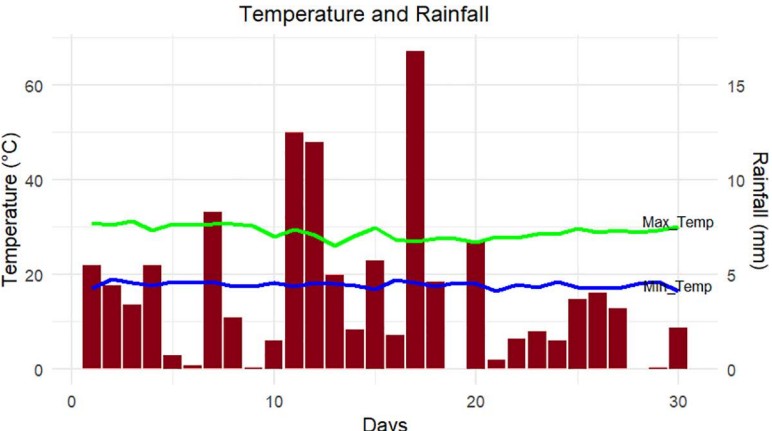

**Fig 2. Temperature and Rainfall prevailing in Jua kali area at the time of mosquito sampling in November 2022 (Source: Kenya Meteorological Department, 2022).**

supernatant was collected and transferred to a different tube for use in cell culture and metagenomics. For metagenomics analyses, 50 µL of supernatant from each individual pool was combined to create four (4) super pools where three super pool had 22 pools and one super pool had 20 pools based on mosquito species and study site.

## Viral isolation

Virus isolation was performed on the 86 pools in Vero cells (African green monkey kidney) (CCL-81™). Briefly, Vero cells were grown overnight at 37°C and 5% $CO_2$ in minimum essential medium supplemented with 2% glutamine, 2% penicillin/streptomycin/amphotericin, 10% fetal bovine serum, and 7.5% $NaHCO_3$ in 24-well plates (Corning, Incorporated). At 80% confluence, a 50 µL aliquot of the clarified supernatant from individual pools was inoculated into each well. The plates were incubated for 1 hour in a humidified incubator at 37°C and 5% $CO_2$ with gentle rocking of the plates every 15 minutes for virus adsorption. Following incubation, 1mL of maintenance medium, comprising minimum essential medium supplemented with 2% glutamine, 2% penicillin/streptomycin/amphotericin, 2% fetal bovine serum, and 7.5% $NaHCO_3$, was added. The cultures were then grown at 37°C and 5% $CO_2$ and monitored daily for cytopathic effects (CPE) for 14 days. Cultures exhibiting CPE were harvested and further passaged by inoculating onto fresh monolayers of Vero cells (CCL 81™) in 25-$cm^3$ cell culture flasks. After two successive passages, the supernatants of virus-infected Vero cell cultures exhibiting cytopathic effect of approximately 70% were harvested from the flasks for virus identification through next-generation sequencing.

## PCR, library preparation and next-generation sequencing

Viral particles from CPE-positive cultures and from super-pools were recovered through 0.22µm filters (Millipore, Merck). From an aliquot of 140 µL of the supernatant, viral RNA was extracted using the QIAamp Viral RNA Mini Kit (Qiagen, Hilden, Germany) and eluted in one step with 60 µL of elution buffer. To confirm the presence of Dengue virus (DENV) in the sample, RT-PCR was performed using RealStar® Dengue RT-PCR Kit 2.0 (Altona Diagnostics, Germany) for further confirmation. The cycle threshold (Ct) value was recorded for the sample followed by direct sequencing on the RNA. Paired-end library for high-throughput sequencing was prepared using Illumina RNA Prep with Enrichment kit (Illumina, USA). The library was prepared following the manufacturer's recommended protocol. The final library was denatured with NaOH and diluted further to 12pM before loading onto the Illumina MiSeq sequencing machine. Sequencing was

performed using MiSeq Reagent V3 (Illumina, USA), in a 300-paired end cycle sequencing format. Negative controls were incorporated in both steps to ensure the validity of the final results.

## Sequence analysis and virus identification

Initial analysis was performed using the CZ-ID platform, an integrated pipeline that offers quality control, de-hosting, duplicate removal, assembly, and viral identification capabilities. After this initial processing, the de-hosted sequence reads were retrieved for further analysis. To validate CZ-ID pipeline results, PrinseqLite v0.20.4 tool was used to filter low-quality reads and remove adapters on command line. De novo sequence assembly was conducted using MEGAHIT v1.2.9 [46] where dengue contigs were recovered and compared to CZ-ID pipeline results. This was followed by mapping the reads back to the generated contigs using BWA [47] to produce SAM files. The subsequent generation of BAM files, along with sorting, indexing, and filtering of contigs that did not meet the specified read mapping thresholds, was handled using Samtools v1.20. Only contigs with an average depth of coverage of ≥10 and a length of ≥500 bp were retained for further analysis. These contigs were first compared against a local version of the NCBI viral database using Diamond v2.0.4 [48]. To ensure specificity, putative viral contigs were further compared to the entire non-redundant protein database (nr), to exclude any non-viral contigs. A stringent e-value threshold of 1e-5 was employed throughout the homology searches to minimize false-positive hits.

## Phylogenetic analysis of the identified RNA viruses

To describe the identified viruses in an evolutionary context, publicly available viruses belonging to these different groups, and more specifically those closely related to the viral strains obtained in the current study were downloaded and used as reference sequences in the reconstruction of phylogenetic trees. RNA-dependent RNA polymerase (RdRp) gene was used to carry out phylogenetic analysis. Closely related RdRp gene sequences were retrieved from UniProt and used as reference sequences in reconstructing the phylogenetic relationship of the viral sequences. The combined set of sequences were codon-aligned using Muscle software embedded in Molecular Evolutionary Genetics Analysis v.7.0 (MEGA7) [49] platform. The aligned sequences were edited using the Bioedit tool and maximum likelihood phylogenetic analysis carried out using IQ-TREE v1.6.12. The best model (GTR+G4 (General Time Reversible+Gamma)) and tree search was performed simultaneously based on 1000 bootstrap estimates and approximate likelihood ratio test.

## Results

### Entomological investigation

A total of 3,649 immature mosquitoes were collected from various water container types, reared to adulthood, and identified to species. Tyres were the most productive breeding sites, accounting for 87.9% of the total immature mosquitoes collected (Table 1). Adult mosquitoes were pooled in 1.5 mL tubes, with a maximum of 25 specimens per pool, and stored at −80°C for viral analysis.

**Table 1. Distribution of wet containers sampled and immatures positivity by container type in Jua Kali, Kisumu.**

| Container Type | Total Containers | Wet -ve | Wet+ve | Total immatures |
|---|---|---|---|---|
| Tyre | 52 | 24 | 28 | 3208 |
| Can/Tin | 4 | 1 | 3 | 269 |
| Jerry can | 2 | 0 | 2 | 172 |
| Total | | | | 3,649 |

## Viral isolation

Of the 86 pools tested one pool (denoted as F67), exhibited clear cytopathic effect (Fig 3A) day 9 post-inoculation. The PCR assay resulted in a Ct value of 33. Next generation sequence analysis of the cell culture isolate resulted in 4 fragments of Dengue virus type-3 with lengths of 726 bp, 318 bp, 312 bp and 290 bp. Phylogenetic analysis revealed the circulating strains in the area as DENV-3 genotype IIIB (Fig 3B).

## Metagenomic analysis of sequencing reads from *Aedes aegypti*

Metagenomic analysis of samples from 4 Superpools revealed a diverse range of viruses (Table 2). Sequencing reads were assigned to three viral families: Flaviviridae, Iflaviridae, and Negeviridae, with one virus remaining unclassified.

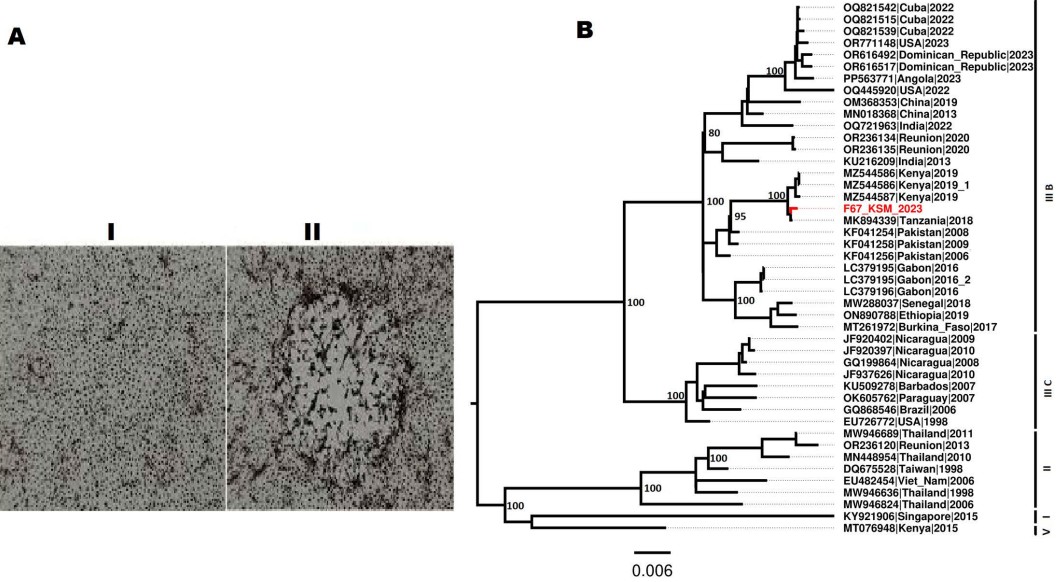

**Fig 3. A. Shows the innoculated cells; I before CPE and II with CPE, B. A Maximum Likelihood phylogenetic tree constructed with 1000 bootstrap replicates using the NS5 protein.** The bootstrap value threshold on the nodes was set at 70.

**Table 2. List of viruses detected during the study, including their identification details based on sequence analysis. The identification was carried out using a homology search against reference databases.**

| Pool Number | Pool size | Name | Length | Closest Hit | GenBank ID | E-value | Identity Score (%) | Query Cover |
|---|---|---|---|---|---|---|---|---|
| F67 | 25 | Flavivirus | 726 | Dengue virus type 3 | MZ544587 | 0 | 99.45% | 100% |
| S04 | 550 | Kisumu Iflavirus 1 | 18858 | Tesano Aedes Virus | LC496784.1 | 0 | 79.71% | 98% |
| | | Unclassified Virus | 3920 | Korle-bu Aedes Virus | LC496785.1 | 0 | 92.64% | 100% |
| | | Kisumu Iflavirus 3 | 4205 | Sassandra virus | MZ202266.1 | 0 | 74.05% | 68% |
| | | Kisumu Iflavirus 2 | 5172 | Armigeres Iflavirus | LC310707.1 | 0 | 74.63% | 100% |
| S05 | 550 | Kisumu Iflavirus 2 | 6943 | Armigeres Iflavirus | LC310707.1 | 0 | 74.39% | 100% |
| | | Kisumu Iflavirus 1 | 7180 | Tesano Aedes Virus | LC496784.1 | 0 | 79.94% | 100% |
| S06 | 500 | Kisumu Iflavirus 1 | 8813 | Tesano Aedes Virus | LC496784.1 | 0 | 79.34% | 100% |
| | | Kisumu Iflavirus 4 | 4470 | Hanko Iflavirus 1 | ON949934.1 | 0 | 76.24% | 66% |
| | | Negevirus | 8005 | Rabai Virus | BK061607.1 | 0 | 99.56% | 99.81% |

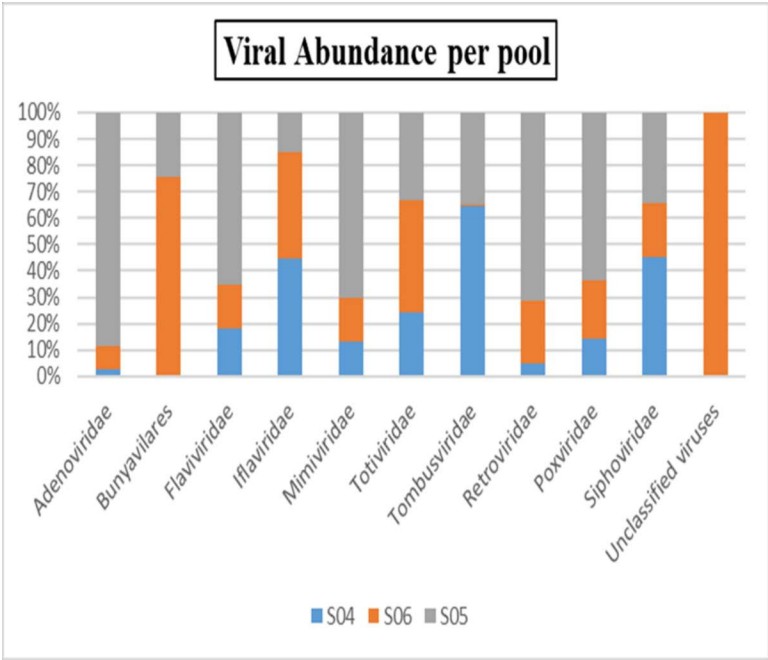

Using DIAMOND the analysis identified contigs corresponding to six distinct viruses. Four Iflaviruses were detected, closely related to Tesano *Aedes* Virus, Armigeres Iflavirus, Sassandra Virus, and Hanko Iflavirus 1. A complete genome of an unclassified virus, closely related to Korle-bu *Aedes* Virus, was obtained. Additionally, a Negevirus closely related to *Rabai* Virus was identified, showing a near-complete genome. Phylogenetic analysis based on highly conserved RdRp domains indicated a close relationship with other viruses from similar metagenomic studies of mosquitoes.

## Taxonomic identification of viruses and their abundances

Taxonomic read classification using the Microbiome Metagenomic pipeline [50] revealed that 90% of the viral reads in the metagenomic pools belonged to the family Iflaviridae, followed by unclassified viruses and Tombusviridae (Fig 4).

## Phylogenetic analysis identified diverse ISVs

The viruses identified in this study were classified into four families: Iflaviridae, Flaviviridae, unclassified viruses, and a yet-to-be-described Negevirus taxon. The identified Negevirus clustered with Negev-like viruses, indicating a close evolutionary relationship with this emerging group. Iflavirus 1 and Iflavirus 2 grouped with other known Iflaviruses, showing high similarity to a diverse array of virus strains, many of which remain unclassified within their respective families or as unclassified RNA viruses. Phylogenetic analysis (Fig 5) using the maximum likelihood method revealed that Iflavirus 1 branched closely with Tesano *Aedes* Virus (Fig 5B). It had a bootstrap value of 89, indicating a strong evolutionary relationship. Additionally, Iflavirus 1 was found to be 79% identical to the only known Tesano Aedes Iflavirus sequence in NCBI, which was first identified in Ghana. Iflavirus 2 clustered with Armigeres Iflavirus, with a bootstrap value of 99, demonstrating a highly supported evolutionary relationship. Iflavirus 2 was found to be 74% identical to the only known protein sequence available. The Unclassified virus formed a clade with Korle-bu Aedes virus, clustering with unclassified mosquito-associated viruses, suggesting potential evolutionary links within this group. Notably, phylogenetic trees for Iflavirus 3

**Fig 4. A hundred percent stacked column chart of relative abundance of the different viral families.**

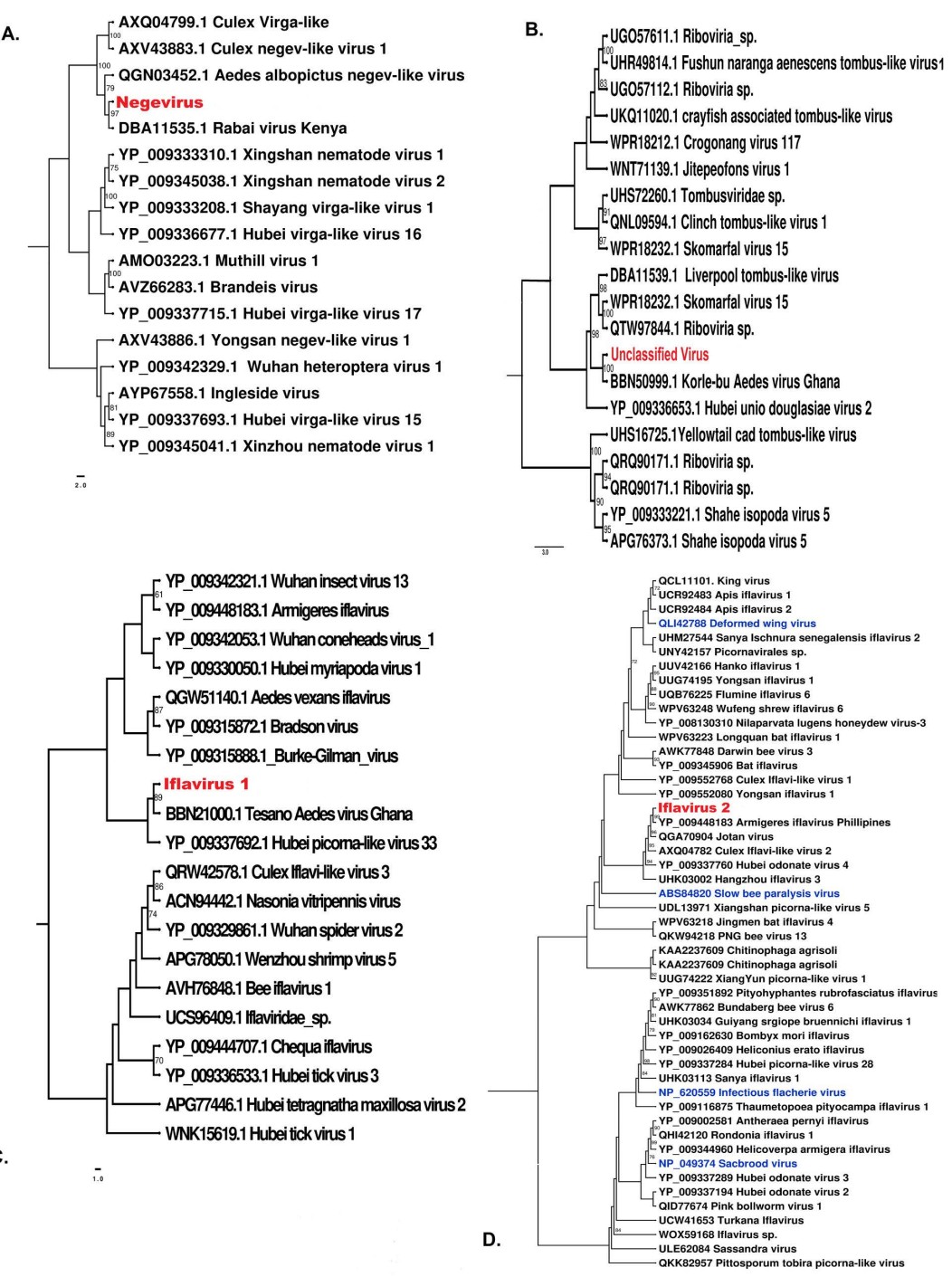

**Fig 5. Maximum likelihood phylogenetic trees constructed using the conserved amino acid domains detected in this study.** A. Negevirus (Accession no. PP590325). B. Unclassified Virus (Accession no. PP590324). C. Iflavirus 1 (Accession no. PP590327). D. Iflavirus 2 (Accession no. PP590328). Bootstrap support values from 1000 bootstrap replicates are indicated on the branches of each tree. Bootstrap values >70% are displayed at important nodes.

and Iflavirus 4, closely related to Sassandra virus and Hanko Iflavirus 1 respectively, were not constructed because the sequences obtained did not have the RdRp segments.

## Discussion

Exploring the mosquito virome and unravelling how its composition influences arbovirus transmission is paramount to understanding the emergence of arboviral diseases and the dynamics of outbreaks.

The present study revealed that the DENV-3 strain identified in Kisumu shares a close phylogenetic relationship with the Tanzanian strain detected in 2018 and the strain that circulated in coastal Kenya in 2019, forming a distinct clade (Fig 3). The close clustering suggests a shared evolutionary history or recent common ancestor among these strains. It also suggests that the Kisumu strain likely originated from the coastal region and has since disseminated into the Kenyan hinterland. The movement of the virus from coastal to inland regions reflects broader trends in the spread of DENV-3, likely driven by both human movement and ecological factors that support the vector, *Aedes aegypti*. Since the Tanzanian strain and the Kenyan 2019 strain are from East Africa, this could indicate regional circulation and potential cross-border spread of this lineage within the area. The clade that includes the Tanzanian, Kenyan 2019, and the newly detected strain branches with dengue strain detected in Pakistan. This suggests that these strains, while geographically separated, are evolutionarily linked and may have a common ancestor at some point in the recent past. This connection could indicate the movement of dengue virus strains across regions, possibly facilitated by travel or trade [51]. This also reflects genetic diversity in circulating strains due to mutation and adaptation, allowing the virus to persist in multiple regions with similar sequences. The presence of closely related dengue strains in Tanzania and Kenya indicate that there might be ongoing circulation of a shared viral lineage across these regions. This emphasizes the need for regional cooperation in surveillance to monitor and control dengue outbreaks. The branching with the Pakistani strain could indicate a historical link between East African and South Asian dengue outbreaks, which may involve shared transmission routes or migration patterns [52]. This highlights the importance of global surveillance for tracking the spread of similar viral lineages. Understanding these evolutionary relationships aids in identifying common mutations and unique adaptations, which can inform vaccine strategies for targeted immunization.

Phylogenetic analysis clustered our sequence correctly within the DENV lineage, supporting its identity. Therefore, detection of the NS5 region was consistent with the natural transmission dynamics of the virus, rather than active replication. This findings have significant implications for understanding dengue virus maintenance during inter-epidemic periods and suggests that vertical transmission might play a role in the virus's persistence in endemic areas. Recent studies have described DENV vertical transmission in wild mosquitoes in different geographical regions, however, research in Kenya indicates that vertical transmission may play a more crucial role [10]. The role of vertical virus transmission has been considered under multiple scenarios. It has been stimulated by theory that transmission of virus through the germ line may provide an alternate mechanism for virus maintenance in nature [33]. Some studies suggest that vertical transmission allows re-emergence and maintenance of arboviruses in a vector population in different scenarios: (i) in between non-epidemic periods; (ii) when the density of mosquito population is low (due to winter diapause); or (iii) in the absence of any viremic hosts [53]. It has also been suggested that virus in diapausing eggs can remain viable until mosquito emergence, thus the virus can multiply and be re-introduced horizontally to the vertebrate host population [54].

The weather conditions in Jua Kali during November 2022, characterized by sporadic but significant rainfall and stable warm temperatures, likely contributed to the proliferation of *Aedes aegypti* mosquitoes and the transmission of Dengue-3 virus. The graph (Fig 2) illustrates several rainfall events, with a notable peak around day 20, which likely led to the accumulation of stagnant water—ideal breeding sites for *Aedes aegypti*, the primary vector of dengue viruses [55]. The relatively stable maximum temperatures around 30°C and minimum temperatures slightly below 20°C may have further facilitated rapid mosquito development, as temperature plays a crucial role in mosquito life cycle progression and virus replication [56].

Female mosquitoes lay eggs in artificial and natural water containers, such as discarded tires, flowerpots, and bottle caps [9]. These eggs can remain viable in a desiccated state for extended periods, hatching when they come into contact with water [57]. Following rainfall, the eggs hatch into larvae, which mature into adults capable of transmitting the dengue virus through bites, thus perpetuating the transmission cycle [35]. The evidence of Dengue-3 vertical transmission during the dry season is particularly significant, as it suggests that the virus can persist in mosquito eggs and larvae even when adult populations decline, posing a risk for sudden outbreaks once favorable environmental conditions return [58].

Negevirus identified in this study (Fig 5A) is closely related to the Rabai virus, initially identified in coastal Kenya [59]. This indicates a strong evolutionary relationship between these two viruses, suggesting they share a recent common ancestor or may have evolved under similar selective pressures. Since Rabai virus was also identified in Kenya, this clustering suggests that similar Negevirus lineages may be circulating across different Kenyan regions, potentially adapting to local environments or vector species. The Negevirus identified formed a clade with Negev-like viruses from *Culex* and *Aedes albopictus* species. This shows a common lineage shared among Negev-like viruses found in multiple mosquito species and are capable of infecting diverse mosquito vectors broadening their potential ecological niche [59]. It also indicates that it might have adapted to various vector species, which could play a role in the persistence and potential spread of Negev-like viruses in diverse environments. Given the potential for a broad host range within mosquito species, monitoring Negeviruses in both *Aedes* and *Culex* populations could be valuable. Emerging evidence suggests that Negeviruses may interact with co-circulating viruses within the same host, potentially modulating the replication or pathogenicity of arboviruses [30]. Studies have shown that Negeviruses, can influence arbovirus transmission by interfering with viral replication in mosquito vectors, a phenomenon known as superinfection exclusion [60]. For example, some Negeviruses have been observed to suppress replication of flaviviruses and alphaviruses in co-infected mosquitoes or cell cultures, reducing the likelihood of secondary infection transmission [61]. Additionally, Negeviruses have been detected in diverse mosquito populations across multiple geographic regions, raising questions about their ecological roles and potential use in biocontrol strategies [59,62]. Their ability to persist in mosquito populations without causing apparent harm suggests they may be leveraged for arbovirus control, either through direct interference with pathogenic viruses or by modulating vector competence.

The unclassified virus detected in this study (Fig 5B) branched with Korlebu Aedes virus signifying a strong evolutionary connection. This indicates that these viruses share a close common ancestor. The fact that Korlebu Aedes virus was originally identified in Ghana while the unclassified virus was detected in Kenya points to a potential trans-African distribution. This indicates either recent movement of similar viral strains across African regions or a wide distribution of this virus family across various Aedes mosquito populations in Africa. The presence of this unclassified virus and Korlebu Aedes virus in *Aedes aegypti* populations suggests it may be particularly adapted to this mosquito species [63]. The unclassified virus formed a sister clade with Tombus-like viruses which are known to infect a range of hosts, including plants and insects [64]. This indicates that it has evolved to occupy a unique ecological niche within mosquito populations. The length of the unclassified virus was 3920nt (Table 2) while that of Korlebu Aedes Virus is 2819nt (Accession number LC496785.1), which indicates distinct functional regions in the unclassified virus thus providing a unique opportunity to delve into potential functional elements that may differentiate these viruses within the Aedes population. Tombus-like viruses belong to the order *Tolivirales* and are closely related to the *Tombusviridae* family, which primarily includes plant-infecting viruses [64]. However, recent metagenomic studies have expanded our understanding of these viruses, revealing their presence in diverse invertebrates, including insects such as mosquitoes, reflecting co-evolution with arthropod hosts [65].

Iflavirus 1 (Fig 5C) detected in this study branched with the Tesano Aedes Virus detected in Ghana which shows a close evolutionary relationship. It formed a sister clade with other iflaviruses from diverse species indicating a shared ancestry and possibly conserved functional domains among these iflaviruses. The length of Iflavirus 1 detected (Table 2) was 18858nt and for Tesano Aedes Virus is 9385nt, this could suggest that Iflavirus 1 has additional coding or regulatory regions. This genomic data strengthens the foundation for further analyses, to understand its potential role in vector

biology and its interactions with other arboviruses.Iflavirus 2 (Fig 5D) from *Aedes aegypti* branched with the Armigeres Iflavirus, which was isolated in the Philippines from *Armigeres* species. This cross-species relationship indicate a broader host range and may share ancestral genetic elements across different mosquito genera, hinting at past interspecies transmissions or shared ecological niches. The presence of pathogenic iflaviruses in sister clades, marked in blue, that infect economically important insects like bees and silkworms, emphasizes potential functional virulence. Further genomic comparison and functional analysis between pathogenic iflaviruses and mosquito-associated iflaviruses could shed light on shared pathogenic mechanisms and reveal any potential impact on the vector species themselves or on arbovirus transmission dynamics. Iflaviruses have been extensively studied in honeybees, where species such as Deformed Wing Virus (DWV) and Sacbrood Virus (SBV) have been linked to colony collapse disorder [66]. In mosquitoes, iflaviruses remain largely uncharacterized, but emerging evidence suggests that they could influence vector biology, immunity, and even interactions with arboviruses [67]. The presence of multiple iflaviruses in *Aedes aegypti*, as observed in this study, raises important questions about their role in vector competence and disease transmission.

This study provides a comprehensive analysis of the virome within *Aedes aegypti* populations in Kisumu, Kenya, identifying a diverse range of viruses, including ISVs and newly detected virus with significant phylogenetic associations. ISVs have been shown to modulate the replication and pathogenicity of co-infecting arboviruses, such as dengue virus potentially altering the transmission dynamics [68]. Additionally, other studies have demonstrated that they can interfere with the replication of West Nile virus (WNV) and Zika virus, influencing overall viral load and transmission rates [69]. Initial in vivo studies with several Culex species suggested that mosquitoes infected with CxFV were less susceptible to secondary WNV infection than control mosquitoes, further highlighting the role of Iflaviruses in modulating susceptibility to other viral infections [24,27,70]. Moreover, Kenya has been a site for the discovery of several other Iflaviruses, including the Mombasa Aedes Iflavirus and Anopheles Iflavirus, found in different mosquito species across the country [10,31]. A study conducted by the Center for Virus Research, KEMRI, also identified diverse Iflaviruses in mosquito populations in coastal and western regions of Kenya, further underscoring the potential role of these viruses in influencing the transmission dynamics of arboviruses [71]. The genetic similarities and differences observed among the ISVs identified in this study could have significant implications for their interactions with arboviruses, highlighting the need for further research into the ecological roles of these viruses.

## Conclusion

The key findings from this study is the evidence of natural vertical transmission, specifically transovarial transmission of DENV-3 genotype III, lineage B and the presence of insect-specific viruses (ISVs) within *Aedes aegypti* populations in Kisumu. Co-evolutionary signals, low genetic variability, and consistent clustering with mosquito lineage confirm this transmission route in the present study. The detection of ISVs adds an additional layer of complexity, as these viruses may influence the vector's susceptibility to dengue and other arboviruses, possibly affecting the transmission dynamics within the local mosquito population. These findings underscore the need for proactive and comprehensive vector control strategies that address not only direct transmission pathways but also consider the ecological interactions between ISVs and arboviruses. By highlighting the potential for intrinsic virus maintenance within vectors, this study contributes valuable insights toward understanding arbovirus epidemiology and improving vector management practices to reduce the risk of future dengue outbreaks. Successful application of metagenomics to characterize viruses, highlights the potential for the application of this method in high-throughput surveillance of viruses. This highlights great promise in the detection of novel viruses and pathogenic viruses, which is useful in prevention of disease outbreaks before they can occur.

## Acknowledgments

We thank Hellen Koka, Victor Ofula, Samson Konongoi, Francis Mulwa, Bryson Kimemia and Jane Thiiru for their expert contribution in cell culture and data analysis. We also thank Samuel Owaka for his contribution to generating the map of sampling sites.

This Material has been reviewed by the Walter Reed Army Institute of Research. There is no objection to its presentation and/or publication. The opinions or assertions contained herein are the private views of the author, and are not to be construed as official, or as reflecting true views of the Department of the Army or the Department of Defense.

## Author contributions

**Conceptualization:** Solomon Langat, Fredrick Eyase.

**Data curation:** Tabitha Wanjiru, Solomon Langat.

**Formal analysis:** Tabitha Wanjiru, Solomon Langat.

**Funding acquisition:** Fredrick Eyase.

**Investigation:** Tabitha Wanjiru, Solomon Langat, Santos Yalwala, David Oullo, Richard Ochieng, Francis Ngere, Gladys Kerich, Janet Ambale, Eunice Achieng, David Abuom, Timothy Egbo, Jaree Johnson, Elly Ojwang, John Eads, Eric Garges.

**Methodology:** Tabitha Wanjiru.

**Project administration:** Timothy Egbo, Elly Ojwang, Eric Garges.

**Resources:** Tabitha Wanjiru, John Eads, Fredrick Eyase.

**Software:** Tabitha Wanjiru, Solomon Langat.

**Supervision:** Wallace Bulimo, Johnson Kinyua, Nicholas Odemba, Santos Yalwala, David Oullo, Richard Ochieng, Francis Ngere, Gladys Kerich, Janet Ambale, Eunice Achieng, David Abuom, Timothy Egbo, Jaree Johnson, Elly Ojwang, John Eads, Eric Garges, Fredrick Eyase.

**Validation:** Tabitha Wanjiru, Fredrick Eyase.

**Visualization:** Tabitha Wanjiru, Solomon Langat.

**Writing – original draft:** Tabitha Wanjiru, Wallace Bulimo, Solomon Langat, Fredrick Eyase.

**Writing – review & editing:** Tabitha Wanjiru, Wallace Bulimo, Solomon Langat, Fredrick Eyase.

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
