## [Decision Letter · Decision Letter 0]

Vertical transmission of Dengue virus type-3 and metagenomic virome profiles of Aedes aegypti mosquitoes collected in Kisumu, Kenya.

PLOS ONE

Dear Dr.  Ng'ang'a ,

Thank you for submitting your manuscript to PLOS ONE. After careful consideration, we feel that it has merit but does not fully meet PLOS ONE’s publication criteria as it currently stands. Therefore, we invite you to submit a revised version of the manuscript that addresses the points raised during the review process.

The manuscript must be revised, the results will improve increasing ovitramps number for the analysis.  Both reviewers have pointed out areas of your paper that need clarification and explanation to improve the relevance of your findings.

Please submit your revised manuscript by May 01 2025 11:59PM. If you will need more time than this to complete your revisions, please reply to this message or contact the journal office at plosone@plos.org . Please include the following items when submitting your revised manuscript:

We look forward to receiving your revised manuscript.

Kind regards,

Victoria Pando-Robles, Ph.D.

Academic Editor

PLOS ONE

Journal Requirements:

“Armed Forces Health Surveillance Branch (AFHSB) and its Global Emerging Infections Surveillance (GEIS) Section, FY2022 ProMIS ID: P0116_22_KY and FY2023 ProMIS ID P0094_23_KY.”

7. Please amend the manuscript submission data (via Edit Submission) to include author Wallace Bulimo, Solomon Langat, Johnson Kinyua, Nicholas Odemba, Santos Yalwala, David Oullo, Richard Ochieng, Francis Ngere, Gladys Kerich, Janet Ambale, Eunice Achieng, David Abuom, Timothy Egbo, Jaree Johnson, Elly Ojwang, John Eads, Eric Garges and Fredrick Eyase.

8. Your ethics statement should only appear in the Methods section of your manuscript. If your ethics statement is written in any section besides the Methods, please move it to the Methods section and delete it from any other section. Please ensure that your ethics statement is included in your manuscript, as the ethics statement entered into the online submission form will not be published alongside your manuscript.

9. We note that Figure 1 in your submission contain map/satellite images which may be copyrighted. All PLOS content is published under the Creative Commons Attribution License (CC BY 4.0), which means that the manuscript, images, and Supporting Information files will be freely available online, and any third party is permitted to access, download, copy, distribute, and use these materials in any way, even commercially, with proper attribution. For these reasons, we cannot publish previously copyrighted maps or satellite images created using proprietary data, such as Google software (Google Maps, Street View, and Earth). For more information, see our copyright guidelines: http://journals.plos.org/plosone/s/licenses-and-copyright.

Reviewers' comments:

Reviewer's Responses to Questions

**Comments to the Author**

1. Is the manuscript technically sound, and do the data support the conclusions?

Reviewer #1: Partly

Reviewer #2: Partly

2. Has the statistical analysis been performed appropriately and rigorously?

Reviewer #1: N/A

Reviewer #2: Yes

3. Have the authors made all data underlying the findings in their manuscript fully available?

Reviewer #1: Yes

Reviewer #2: Yes

4. Is the manuscript presented in an intelligible fashion and written in standard English?

Reviewer #1: Yes

Reviewer #2: Yes

Reviewer #1: The study is very intersting, just have the time to review the different points to make your manuscript more attractive. In your study you did not use a second method to confirme the dengue virus strain and also the use of cell culture may have mask other viruses that are not so competent to be amplified in cell culture isolation which you wanted to reveal with NGS but here also the use of 550 samples is too high and could also mask the viruses which are in low quantity.

Reviewer #2: Dear Authors,

Your article is very good; however, I have few major and minor comments that could improve the quality and readability of your article.

Detail comments are attached and try to modify your manuscripts based on comments

**Do you want your identity to be public for this peer review?** For information about this choice, including consent withdrawal, please see our Privacy Policy

Reviewer #1: No

Reviewer #2: No

---

## [Author Response · Author response to Decision Letter 1]

5 Apr 2025

Revision 1

Response 1: The manuscript has been formatted according to PLOS ONE’s styling requirements.

Response 2: Ethical approval was obtained from the Kenya Medical Research Institute (KEMRI) Scientific and Ethics Review Unit (SERU) under protocol number KEMRI/SERU/CCR/4702 and WRAIR# 3101. Permission to conduct the study was granted by the National Council for Science, Technology, and Innovation (NACOSTI).

Response 3: Funding information has been removed from the manuscript

Response 4: This are the correct grant numbers. ‘This work was funded by the Armed Forces Health Surveillance Branch (AFHSB) and its Global Emerging Infections Surveillance (GEIS) Section, FY2022 ProMIS ID: P0116_22_KY and FY2023 ProMIS ID P0094_23_KY.’

“Armed Forces Health Surveillance Branch (AFHSB) and its Global Emerging Infections Surveillance (GEIS) Section, FY2022 ProMIS ID: P0116_22_KY and FY2023 ProMIS ID P0094_23_KY.”

Response 3: The funders had a role on validation of the project, administration and decision to publish

Response 6: The authors have agreed have a data sharing plan before acceptance

7. Please amend the manuscript submission data (via Edit Submission) to include author Wallace Bulimo, Solomon Langat, Johnson Kinyua, Nicholas Odemba, Santos Yalwala, David Oullo, Richard Ochieng, Francis Ngere, Gladys Kerich, Janet Ambale, Eunice Achieng, David Abuom, Timothy Egbo, Jaree Johnson, Elly Ojwang, John Eads, Eric Garges and Fredrick Eyase.

Response 7: All authors have been included on the manuscript submission data

8. Your ethics statement should only appear in the Methods section of your manuscript. If your ethics statement is written in any section besides the Methods, please move it to the Methods section and delete it from any other section. Please ensure that your ethics statement is included in your manuscript, as the ethics statement entered into the online submission form will not be published alongside your manuscript.

Response 8: Ethics statement have been moved to methods section

9. We note that Figure 1 in your submission contain map/satellite images which may be copyrighted. All PLOS content is published under the Creative Commons Attribution License (CC BY 4.0), which means that the manuscript, images, and Supporting Information files will be freely available online, and any third party is permitted to access, download, copy, distribute, and use these materials in any way, even commercially, with proper attribution. For these reasons, we cannot publish previously copyrighted maps or satellite images created using proprietary data, such as Google software (Google Maps, Street View, and Earth). For more information, see our copyright guidelines: http://journals.plos.org/plosone/s/licenses-and-copyright.

Reviewers' comments:

Reviewer's Responses to Questions

Response 9: This is the caption with granted permission. ‘Fig 1. The overall map of Kenya and Kisumu County showing the site where sampling was conducted. Base maps, boundaries and shape files of Kenyan map and administrative boundaries of the County and Sub-county were derived from GADM data version 4.1 (https://gadm.org) and the maps were generated using ArcGIS Version 10.2.2 (http://desktop.arcgis.com/en/arcmap) advanced license) courtesy of Samuel Owaka.

Comments to the Author

1. Is the manuscript technically sound, and do the data support the conclusions?

Reviewer #1: Partly

Reviewer #2: Partly

2. Has the statistical analysis been performed appropriately and rigorously?

Reviewer #1: N/A

Reviewer #2: Yes

3. Have the authors made all data underlying the findings in their manuscript fully available?

Reviewer #1: Yes

Reviewer #2: Yes

4. Is the manuscript presented in an intelligible fashion and written in standard English?

Reviewer #1: Yes

Reviewer #2: Yes

5. Review Comments to the Author

Reviewer #1: The study is very intersting, just have the time to review the different points to make your manuscript more attractive. In your study you did not use a second method to confirme the dengue virus strain and also the use of cell culture may have mask other viruses that are not so competent to be amplified in cell culture isolation which you wanted to reveal with NGS but here also the use of 550 samples is too high and could also mask the viruses which are in low quantity.

Reviewer #2: Dear Authors,

Your article is very good; however, I have few major and minor comments that could improve the quality and readability of your article.

Detail comments are attached and try to modify your manuscripts based on comments

6. PLOS authors have the option to publish the peer review history of their article (what does this mean?). If published, this will include your full peer review and any attached files.

Do you want your identity to be public for this peer review? For information about this choice, including consent withdrawal, please see our Privacy Policy.

Reviewer #1: No

Reviewer #2: No

Responses to reviewers

The authors report in their article the vertical transmission of dengue virus and other insect specific viruses in Aedes aegypti collected in Kisumu, Kenya. The use of cell culture for virus isolation and NGS technique to detect the virome were interesting as approach but it remains some gaps to fulfil.

Abstract

Line 34: change “processed for” by “analysed using”

Response: Line 38 this has been addressed to ‘analysed using’

Line 78-85: please add the number of dengue cases

Response: Line 79-89 this has been addressed and the number of dengue cases have been added

Line 86: you have to write complete name before using abbreviation Insect Specific Viruses (ISVs);

Response: Line 90 complete name has been added

Line 134-135: repetition of the idea, please delete it.

Response: Line 137-139. This has been deleted

Line 169-170: “Larvae and pupae were collected” is better

Response: Line 192-193. This has been corrected

Line 174: the samples were georeferenced

Response: Line 197: This has been addressed

Line 194: adults were completely crushed or a part of the them?

Response: Line 220: yes completely crushed

Comparison between pools and superpools

Response: Line 228-230.

Line 257; which model have you used for your calculation.

Response: Line 287-288. GTR+G4 (General Time Reversible + Gamma)

Line 261: move weather data to material & methods chapter where you describe the study area

Response: Line 180-190. This has moved to methods

Line 271-276: reformulate, detail your results and give significant of the abbreviation in the table 1

Response: Line 296-299.This has been addressed

Results

Miss of the table containing mosquito species identification

Response: The table is missing because the morphological identification had already been done by the entomologists and determined to be Aedes aegypti species which was then processed for analysis.

Line 281: change the title from dengue to viral isolation (follow your M&M plan) and give more details about your results: day of CPE, …

Response: Line 304. This has been addressed

Table 2 coverage rate is not reported which is important also.

Response: Line 327-328 Query cover has been added on the last column to show the coverage rate

Have you amplified NS5 protein from the isolate? It is not reported in M&M for dengue classification? And what is its bootstraps value? What is the threshold used in the tree for nodes value bootstrap?

Response:

• Line 251-254, 306: PCR section has been added in methodology and results

• Fig 3B. bootstrap value is 100 and the threshold for tree nodes is 70

Why did you not perform a second qRT-PCR or end point Rt-PCR to confirm you dengue virus? Because 726bp is not enough and represent only 7% of the total genome

Response

• As stated above we performed qRT-PCR prior to sequencing.

• Line 405- 407: The concern of the reviewer has been addressed by the additional information in

---

## [Decision Letter · Decision Letter 1]

Dear Dr. Ng'ang'a,

We look forward to receiving your revised manuscript.

Kind regards,

Victoria Pando-Robles, Ph.D.

Academic Editor

PLOS ONE

Journal Requirements:

Additional Editor Comments:

Dear authors

The manuscript must explain the calculation of the entomological indices. A minimum of 100 houses should be inspected, but you only describe five collections. You need to correct this discrepancy.

Regards

Reviewers' comments:

Reviewer's Responses to Questions

**Comments to the Author**

Reviewer #2: (No Response)

2. Is the manuscript technically sound, and do the data support the conclusions?

Reviewer #2: Partly

3. Has the statistical analysis been performed appropriately and rigorously?

Reviewer #2: Yes

4. Have the authors made all data underlying the findings in their manuscript fully available?

Reviewer #2: Yes

5. Is the manuscript presented in an intelligible fashion and written in standard English?

Reviewer #2: Yes

Reviewer #2: Dear Authors, Thank you for considering my comments. However one important comment is not addressed. this is not only about the quality and readability of your article but also for readers. Like reference you used to classify entomological indices, different scholars will use your article and five house holds are enough to estimate entomological indices.

**Do you want your identity to be public for this peer review?** For information about this choice, including consent withdrawal, please see our Privacy Policy

Reviewer #2: No

---

## [Author Response · Author response to Decision Letter 2]

10 May 2025

Journal Requirements:

Response

Thank you for your comment. I have carefully reviewed the entire reference list, and I confirm that all cited references are accurate, up-to-date, and none have been retracted. Additionally, I have made a revision to Reference No. 41 (Line 147) to reflect a more current and relevant citation. The updated reference has been included in the revised manuscript accordingly.

Additional Editor Comments:

Dear authors

The manuscript must explain the calculation of the entomological indices. A minimum of 100 houses should be inspected, but you only describe five collections. You need to correct this discrepancy.

Response

Line 203, 293 & 372: This section (Larval Indices) has been removed in Methodology, Results and discussion section.

6. Review Comments to the Author

Reviewer #2: Dear Authors, Thank you for considering my comments. However one important comment is not addressed. this is not only about the quality and readability of your article but also for readers. Like reference you used to classify entomological indices, different scholars will use your article and five house holds are enough to estimate entomological indices.

Response

Line 203, 293 & 372: This section (Larval Indices) has been removed in Methodology, Results and discussion section.

---

## [Decision Letter · Decision Letter 2]

Vertical transmission of Dengue virus type-3 and metagenomic virome profiles of Aedes aegypti mosquitoes collected in Kisumu, Kenya.

PONE-D-24-54015R2

Dear Dr. Tabitha Wanjiru Ng'ang'a,

We’re pleased to inform you that your manuscript has been judged scientifically suitable for publication and will be formally accepted for publication once it meets all outstanding technical requirements.

Kind regards,

Victoria Pando-Robles, Ph.D.

Academic Editor

PLOS ONE

Additional Editor Comments (optional):

Reviewers' comments:

Reviewer's Responses to Questions

**Comments to the Author**

Reviewer #2: All comments have been addressed

2. Is the manuscript technically sound, and do the data support the conclusions?

Reviewer #2: Yes

3. Has the statistical analysis been performed appropriately and rigorously?

Reviewer #2: Yes

4. Have the authors made all data underlying the findings in their manuscript fully available?

Reviewer #2: Yes

5. Is the manuscript presented in an intelligible fashion and written in standard English?

Reviewer #2: Yes

Reviewer #2: (No Response)

**Do you want your identity to be public for this peer review?** For information about this choice, including consent withdrawal, please see our Privacy Policy

Reviewer #2: No

---

## [Editor Report · Acceptance letter]

PONE-D-24-54015R2

PLOS ONE

Dear Dr. Ng'ang'a,

I'm pleased to inform you that your manuscript has been deemed suitable for publication in PLOS ONE. Congratulations! Your manuscript is now being handed over to our production team.

Kind regards,

on behalf of

Dr. Victoria Pando-Robles

Academic Editor

PLOS ONE